# In Situ Grown Vertically Oriented Graphene Coating on Copper by Plasma-Enhanced CVD to Form Superhydrophobic Surface and Effectively Protect Corrosion

**DOI:** 10.3390/nano12183202

**Published:** 2022-09-15

**Authors:** Xiaohang Zheng, Yaqian Yang, Yi Xian, Heng Chen, Wei Cai

**Affiliations:** Harbin Institute of Technology, School of Materials Science and Engineering, Harbin 150001, China

**Keywords:** vertically-oriented graphene, PECVD, super hydrophobic, copper, corrosion resistance

## Abstract

Graphene exhibits great potential for the corrosion protection of metals, because of its low permeability and high chemical stability. To enhance the anticorrosion ability of Cu, we use plasma-enhanced chemical vapor deposition (PECVD) to prepare a vertically oriented few-layer graphene (VFG) coating on the surface of Cu. The Cu coated with VFG shows superhydrophobic surface with a contact angle of ~150°. The VFG coating is used to significantly increase the anticorrosion ability, enhanced by the chemical stability and the unique geometric structure of vertically oriented graphene. The corrosion rate of VFG-Cu was about two orders of magnitude lower than that of bare Cu. This work highlights the special synthesized way of PECVD and superhydrophobic surface of vertical structures of graphene as coatings for various applications.

## 1. Introduction

Cu is widely used in the electronics industry owing to its excellent thermal and electrical conductivities. However, the corrosion of Cu causes the degradation of electronic and thermal properties, which can lead to significant financial costs and safety problems [1,2,3,4]. For example, the lacking corrosion resistance of Cu will cause the degradation of fuel cell stack performance, which is used as a metal bipolar plate of fuel cell. Enhancing the corrosion inhibition of Cu is an important issue. Considering that metallic corrosion is an electrochemical process at the surface, a physical obstacle such as films or coatings is a useful way to reinforce the surface chemical resistance of copper [5,6,7].

Graphene, which comprises a single-atom-thick crystal of sp^2^-bonded carbon, has been seen as a good candidate for corrosion protection, which possesses chemical stabilities and highly flexible [8,9,10,11,12,13]. Qiu et al. [14]. synthesized graphene oxide (GO) with polyaniline for coating on 316 stainless steels. Electrochemical measurements indicated good corrosion inhibition efficiency. Nayak et al. [15]. used 4-nitroaniline and GO nanocomposites to coat on the surface of steel. The nanocomposite coating increased the barrier properties by increasing the required path length of corrosive electrolytes to reach the metal-coating interface. However, most polymers suffer from low thermal stability, so the polymer-modified GO coating is unsuitable for high temperature applications. Additionally, the relatively high porosity and weak adhesion of the composite coating decreases the long-term corrosion resistance. The in situ growth of graphene on metallic substrates has been reported for protective coatings [16,17]. High quality in situ grown graphene strongly adheres to the substrate, which results in better anticorrosion properties. Tafel analysis has revealed that the corrosion rate of multilayer graphene films grown on Ni to be 20 times slower than that of bare Ni, and the corrosion rate of mechanically transferred graphene to be 4 times slower than that of bare Ni [7]. While in situ grown graphene is generally high quality, the defects and grain boundaries are inevitable, which are detrimental to the long-term anticorrosion properties.

In recent years, superhydrophobic coatings fabricated as anticorrosive layers have been applied to metallic substrates [18,19,20]. Arukalam et al. [21]. designed a nanostructured superhydrophobic polysiloxane-ZnO coating for high barrier marine topcoats. Xu et al. [20]. prepared a superhydrophobic surface on Mg alloy that was electrochemically machined and subsequently covered with a fluoroalkylsilane film. The superhydrophobic Mg alloy surface reportedly exhibited good resistance to corrosive liquids during electrochemical testing. The hydrophobic structure on the surface can effectively avoid direct contact between the substrate and corrosive ions. The wettability of a surface depends on its chemical composition and geometric structure [22]. Thus, the anticorrosion properties should be enhanced if a graphene coating on a Cu surface possesses a superhydrophobic character and covers the defects and boundaries.

In the current study, we designed and prepared vertically oriented few-layer graphene coated on Cu (VFG-Cu) by a simple and useful way of plasma-enhanced chemical vapor deposition (PECVD). The VFG-Cu with superhydrophobic properties could effectively improve the corrosion resistance of Cu. The microstructure and electrochemistry corrosion behavior of VFG-Cu and generally synthesized in-plane graphene on Cu (GS-Cu) have been investigated. The VFG coating on Cu exhibited a corrosion inhibition efficiency of 98% and excellent long-term anticorrosion properties, which were superior to those of the bare Cu and GS-Cu surfaces.

## 2. Experimental

### 2.1. Fabrication and Characterization

VFG-Cu was synthesized in situ on the surface of 0.1 mm thick Cu foil by radio-frequency (RF) plasma-enhanced chemical vapor deposition (PECVD) at 700 °C in a mixture of CH_4_/Ar/H_2_ (gas flow rate ratio of 50 sccm/50 sccm/10 sccm, respectively, where sccm denotes standard cubic centimeters per minute at standard temperature and pressure). The total gas pressure was fixed at 600 Pa. The RF power was 200 W, and the growth time was 1 h. After PECVD growth, the system was allowed to cool to room temperature under the H_2_/Ar environment atmosphere.

In-plane GS-Cu was grown in situ on the surface of Cu foil by chemical vapor deposition (CVD). Cu foils were inserted into a 2-inch-diameter quartz tube, and the quartz tube was then placed inside a horizontal furnace. The furnace temperature was ramped from room temperature up to 1000 °C under an H_2_/Ar atmosphere (17% H_2_, gas flow rate: 90 sccm). After reaching the target temperature, the growth of GS was initiated by introducing CH_4_ (gas flow rate: 10 sccm) into the reaction tube at ambient pressure with 10 min. When CVD growth had finished, the system was cooled to room temperature at a cooling rate of 10 °C per min under the H_2_/Ar atmosphere.

The morphologies and structures of the samples were investigated by Scanning electron microscopy (SEM, Quanta600FEG), transmission electron microscopy (TEM, Tecnai G2F30)), and Raman spectroscopy (BWS435-532SY).

### 2.2. Water Contact Angle Measurement

The contact angles of the samples with the electrolytes were measured by using a fully computer-controlled apparatus Data Physics OCA20 contact angle meter system. The contact angles of water droplets on the samples are measured by standing a droplet of deionized water (2 μL) on the sample surface.

### 2.3. Electrochemical Corrosion Test

Open circuit potential (OCP), potentiodynamic sweep (PDS), and electrochemical impedance spectrum (EIS) measurements were conducted in 3.5% NaCl solution in a three-electrode polytetrafluorethylene cell using an electrochemical workstation (CHI-760E and PARSTAT 4000A). The working electrode was the prepared sample (active area: 1 cm^2^), while a Pt film (2 cm × 1 cm) and a saturated calomel electrode (SCE) served as the counter and reference electrodes, respectively. The PDS measurements were carried out from 150 mV to +150 mV with respect to *E*_ocp_ with a scan rate of 1 mV/s. The corrosion potential (*E*_corr_) and the corrosion current density (*j*_corr_) are obtained by Tafel extrapolation technique from polarization curve.

## 3. Results and Discussion

### 3.1. Microstructure of GS and VFG

The as-grown GS were transferred from the Cu substrate to a SiO_2_ (90-nm-thick)/Si substrate using the polymethyl methacrylate (PMMA)-assisted etching method [23,24]. As shown in Figure 1a, after the PMMA layer was carefully removed using acetone, a continuous and interconnected GS was obtained. The large low contrast area suggested that the obtained in-plane GS was thin and continuous. In Figure 1b,c, the structural characteristics of the in-plane GS were investigated by Raman spectroscopy. Three bands were observed at ~1350 cm^−1^ (D band), ~1580 cm^−1^ (G band), and ~2700 cm^−1^ (2D band) [24]. The intensity ratio of the 2D band to G band (I_2D_/I_G_) indicates the quality and layer number of GS, while the intensity ratio of the D band to G band (I_D_/I_G_) is related to the degree of disorder and defects in graphene [25]. The weak D band and extremely low I_D_/I_G_ of ~0.03 indicated the formation of highly ordered graphitic carbon with few structural defects. The 2D band was sharp and symmetric with a full-width at half-maximum of ~50 cm. The I_2D_/I_G_ values were 1.2 and 2.3, which corresponded to the areas marked A and B, respectively. The Raman results suggested that the obtained GS structure was few-layer graphene with an average thickness varying from 1 to 3 atomic layers [24,25,26]. The HRTEM image (Figure 1d) shows that the edge of the GS was mainly composed of 1–3 atomic layers with an interlayer spacing of ~0.34 nm, which is the same as that of the graphite lattice. The Raman results were consistent with the TEM results, indicating that a high-quality graphene film had been fabricated.

As shown in Figure 2, SEM, TEM, and HRTEM images show that the VFG on the substrate had densely exposed edge planes and random expansive open areas. From TEM images (Figure 2b,d), the average height of the VFG nanosheets was estimated to be ~100 nm. Figure 2c shows that the VFG nanosheets were well connected to each other to form an intact and continuous entity. The HRTEM image (Figure 2e) shows a typical edge of the VFG. This revealed the graphite structure of the VFG with an interplanar spacing of 0.34 nm. The edge of the VFG was about 3–5 graphene layers thick. In Raman spectra of the VFG (Figure 2f), the *I_D_*/*I_G_* ratio of the VFG was about ~0.8, indicating that this VFG sample was few-layer graphene with an average thickness of 3–10 atomic layers [25,26,27]. This result was consistent with the HRTEM result. Previous reports have indicated that the in situ growth of VFG by PECVD can be divided into two steps. In-plane graphene first grows on the substrate, and then vertically oriented graphene preferentially nucleates and grows upwards from the defects and boundaries of the in-plane graphene [27,28,29,30,31]. Thus, VFG-Cu is thought to be formed from a layer of vertically oriented graphene on top of the GS-Cu. The above analysis suggested that the vertically oriented graphene grown on Cu had good continuity, few surface defects, and high crystallinity.

### 3.2. Wettability of GS-Cu and VFG-Cu

We then characterized the wettability of the Cu, GS-Cu, and VFG-Cu surfaces towards water. As shown in Figure 3, the contact angles of bare Cu and GS-Cu were ~47° and ~85°, respectively. Pure GS exhibits hydrophobicity to its surface chemical properties, and the contact angle of GS with a planar structure is reportedly 80–90° [32,33]. The current result was similar to previous reports in which the contact angle increased remarkably after the growth of pure high-quality graphene with a planar structure on the Cu surface. The contact angle of Cu with VFG coating was observed to be in superhydrophobic range (~150°), which is far greater than the reported results [34]. The VFG-Cu can be regarded as a large number of GS arrays vertically arranged on the substrate, which could further enhance the hydrophobic state of the surface.

### 3.3. Electrochemical Corrosion Behavior of GS-Cu and VFG-Cu

Figure 4a shows OCP curves of the bare Cu, GS-Cu, and VFG-Cu substrates measured in NaCl solution. The *E*_OCP_ of GS-Cu was shifted to more positive potential compared with that of bare Cu. The VFG-Cu substrate exhibited the highest *E*_OCP_ among the three samples. Anodic and cathodic polarization curves (Figure 4b) for the corrosion of the bare Cu, GS-Cu, and VFG-Cu substrates exposure to NaCl solution.

Corrosion parameters obtained from PDS measurements are listed in Appendix A. There was a significant shift in *E*_corr_ for the GS-Cu substrate to more positive values (−209 mV) compared with that for bare Cu (−289 mV). The *E*_corr_ of the VFG-Cu showed the highest positive value (−125 mV). The *j*_corr_ continuously decreased from bare Cu to VFG-Cu. The corrosion rate (ν_corr_) is calculated according to Equation (1):ν_corr_ = E_w_ × j_corr_/ρF(1)
where the equivalent weight E_w_ is 31.7 g for Cu, the density ρ, is 8.94 g/cm^3^ for Cu, F is the Faraday’s constant (96.483 C/mol), and the sample area A is 1 cm^2^. The ν_corr_ for VFG-Cu was significantly lower than those for the bare Cu and VFG-Cu substrates. The results show clearly that the VFG coating reduces the ν_corr_ from 7.57 × 10^−2^ to 9.14 × 10^−4^ mm/year.

The corrosion inhibition efficiency (*η*) of graphene-coated Cu was estimated using Equation (2):*η* = (*j*_bare Cu_ − *j*_graphene-coated Cu_)/*j*_bare Cu_ × 100%(2)

The corrosion inhibition efficiencies of GS-Cu and VFG-Cu were 88.3% and 98.7%, respectively. This shows the VFG coating was an effective corrosion barrier for the Cu substrate. The anticorrosion efficiency of VFG-Cu is higher than the reported result [35].

The electrochemical properties of the graphene coatings were also evaluated by EIS. The Nyquist plots for the GS-Cu and VFG-Cu (Figure 5a) were shown a single incomplete capacitive semicircle and different diameters. The diameter for the VFG-Cu substrates was larger than that for GS-Cu, indicating the surface resistance offered by VFG coated by Cu was higher than that of bare Cu and GS-Cu. Figure 5b,c show bode plots for the Cu, GS-Cu, and VFG-Cu substrates. Each sample showed only one time constant in the bode plot, but the maximum phase angle increased after coating with graphene. The impedance values exhibited a trend of VFG-Cu > GS-Cu > Cu, at low frequencies. This indicated that the in situ grown graphene on Cu provided good protection, which led to inhibition of the corrosion process.

### 3.4. Morphologies after Long-Term Corrosion

Figure 6 shows the morphologies of the Cu, GS-Cu, and VFG-Cu substrates before and after exposure to 3.5% NaCl solution for 1 month, and a schematic of the anticorrosion mechanism. The entire surface of the bare Cu substrate was damaged after 1 month of exposure and was completely covered by corrosion products. The surface of the GS-Cu was partly degraded and contained fewer corrosion products. There were corrosion points at defects and grain boundaries after 1 month. This indicated that the growth of in-plane graphene on Cu provided some corrosion protection, but that its long-time anticorrosion properties were insufficient, similar to the previous reports [36]. The surface morphology of the VFG-Cu was unchanged after exposure for 1 month, showing no corrosion products or corrosion pitting on the surface. To further characterize their chemical states and corrosion, the EIS measures were used for their stability. From Appendix A, the impendence of VFG-Cu had little changed and that of Cu had obviously decreased, which implies the best anticorrosion ability of VFG-Cu.

### 3.5. Corrosion Protection Mechanism

To reveal the mechanism of the VFG anticorrosion ability, we schematically elucidated the relationship between the graphene film structure and wettability and anticorrosion properties. Figure 6 shows that Cu was readily corroded after exposure to NaCl solution. GS could inhibit the corrosion of Cu due to its good chemical stability and impermeability. However, limitations in the structure and preparation process resulted in many defects and grain boundaries at which long-term corrosion occurred. VFG overcame the deficiencies of GS using its vertical-oriented structure and growth mechanism. The superhydrophobicity of VFG-Cu avoided contact between the corrosive solution and GS on the Cu surface, which further improved the corrosion resistance. Thus, the VFG coating exhibited excellent long-term anticorrosion ability, which was far better than those of bare Cu and GS-Cu.

The significant improvement of the anticorrosion properties of VFG-Cu has been discussed as follows. First, the in situ growth of the low-porosity VFG resulted in strong bonding to the Cu substrate. Thus, the VFG had good mechanical stability, chemical stability, and thermal conductivity. Second, the VFG possessed all the advantageous properties of GS. The VFG preferentially nucleated at defects and boundaries of the in-plane graphene during growth. This somewhat covered the defects of in-plane graphene and overcame its poor long-term anticorrosion properties. Third, the 3-D graphite crystal structure of VFG with good hydrophobic characteristics further enhanced the anticorrosion property. The preparation of VFG coated on Cu is simple and could be readily industrialized. Therefore, VFG is an ideal anticorrosion coating for Cu.

## 4. Conclusions

We have developed a corrosion protection coating based on vertically oriented graphene by PECVD. The VFG coating consisted of in-plane graphene similar to GS but also contained a graphene array vertically oriented on the surface of the in-plane graphene. The VFG-Cu substrate showed excellent hydrophobicity, with a water CA of ~150° compared with ~47° and ~85° for the bare Cu and GS-Cu substrates, respectively. Electrochemical test results confirm that VFG coating dramatically increases the resistance of Cu to electrochemical corrosion. As for the protective effect of graphene and special hydrophobicity structure, the corrosion inhibition efficiency of VFG-Cu is proved to be 98.7% in a severe chloride environment. The VFG coating also shows great potential for long-time anticorrosion, since the VFG coated Cu is undamaged even in a very harsh environment (3.5% NaCl) for one month. The special and useful vertically oriented graphene structure via PECVD to form a superhydrophobic surface may pave the way toward the development of high performance for anticorrosion coatings and other applications.

## Figures and Tables

**Figure 1 nanomaterials-12-03202-f001:**
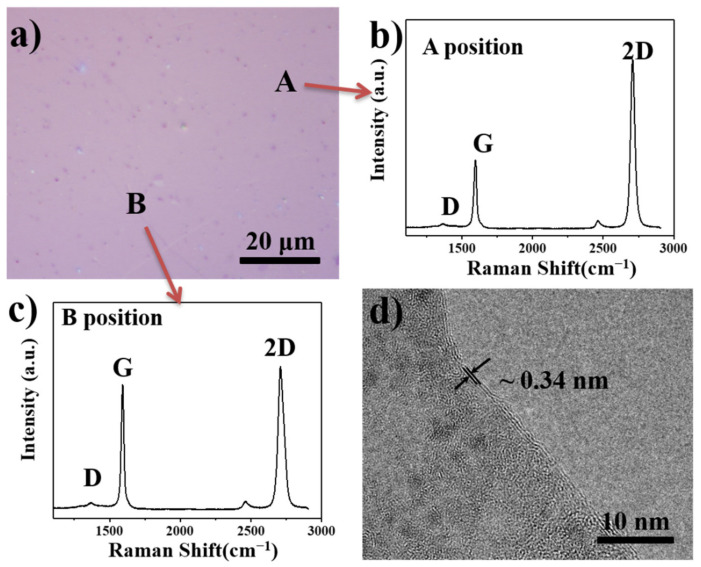
(**a**) SEM image of GS, Raman spectra of the area marked (**b**) A and (**c**) B in (**a**). and (**d**) HRTEM image of GS.

**Figure 2 nanomaterials-12-03202-f002:**
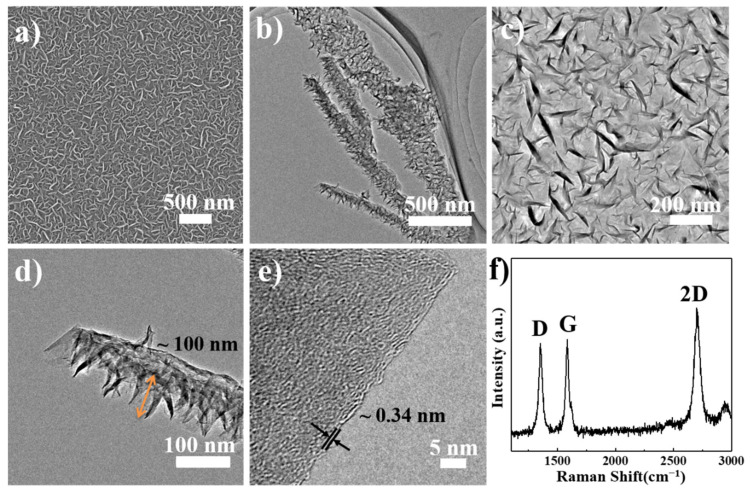
(**a**) SEM image of VFG, (**b**–**d**) TEM images of VFG, (**e**) HRTEM image of VFG, and (**f**) Raman spectra of VFG.

**Figure 3 nanomaterials-12-03202-f003:**
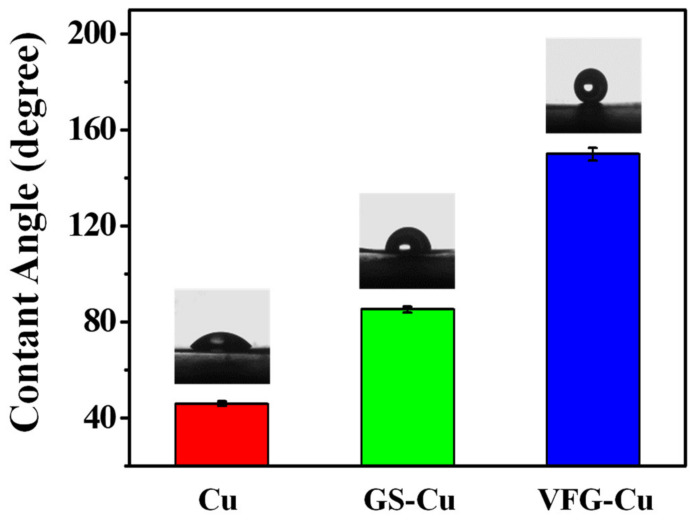
Contact angles of bare Cu, GS-Cu and VFG-Cu.

**Figure 4 nanomaterials-12-03202-f004:**
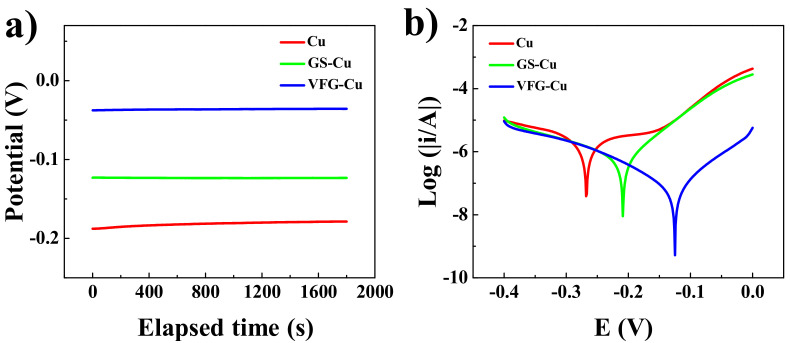
(**a**) Open circuit potential curves for Cu, GS-Cu, and VFG-Cu. (**b**) Tafel plots for Cu, GS-Cu, and VFG-Cu.

**Figure 5 nanomaterials-12-03202-f005:**
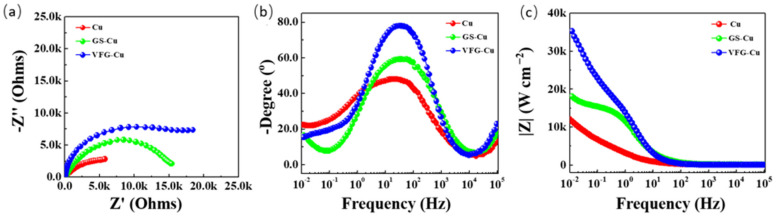
(**a**) Nyquist plots for Cu, GS-Cu, and VFG-Cu (**b**), (**c**) Bode magnitude plots for Cu, GS-Cu, and VFG-Cu.

**Figure 6 nanomaterials-12-03202-f006:**
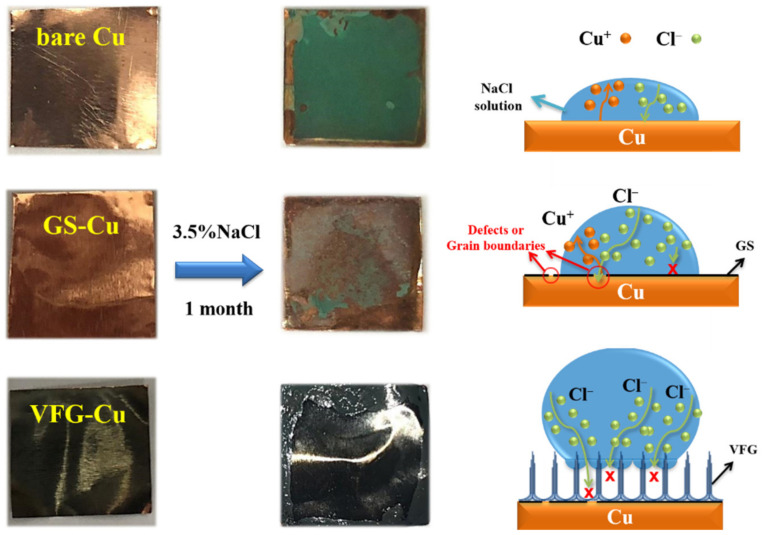
Morphologies of Cu, GS-Cu, and VFG-Cu before and after exposure to aqueous 3.5% NaCl for 1 month, and schematic of the anticorrosion mechanism.

## Data Availability

The raw/processed data required to reproduce the current findings cannot be publicly available at this time as the data also forms part of an ongoing study.

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
