# Peer review of "In Situ Grown Vertically Oriented Graphene Coating on Copper by Plasma-Enhanced CVD to Form Superhydrophobic Surface and Effectively Protect Corrosion"

_nanomaterials, 2022, doi:10.3390/nano12183202_

Round 1

Reviewer 1 Report

The work presented in this manuscript is interesting and worthy to be published. However, there are some minor issues have to be addressed before acceptance as listed.

1. English need to be polished.

2. Abstract need to be re-write highlight what is new and achivements in numbers.

3. Introduction need to be enriched with with graphene produced with other green roustes and I recommed citation of the following reports

Materials Letters 158 (2015) 186-189

Journal of Polymers and the Environment 26 (2018) 1072-1077

4. Comparative study need to be conducted with reported results.

5. Conclusion need to enriched reason of anticorrosion actions.

Author Response

Response to Reviewer 1 Comments

  1. English need to be polished.

Response 1: Thanks for reminding. Based on your comment, we have improved this language of our manuscript.

  1. Abstract need to be re-write highlight what is new and achivements in numbers.

Response 2: Thanks very much for your advice. Following your suggestion, we have re-write the abstract and highlight its new and achievements.

Abstract: Graphene exhibits great potential for the corrosion protection of metals, because of its low permeability and high chemical stability. To enhance the anticorrosion ability of Cu, we use plasma-enhanced chemical vapor deposition (PECVD) to prepare a vertically oriented few-layer graphene (VFG) coating on the surface of Cu. The Cu coated with VFG shows superhydrophobic surface with a contact angle of ~150°. The VFG coating is used to significantly increase the anticorrosion ability, enhanced by the chemical stability and the unique geometric structure of vertically oriented graphene. The corrosion rate of VFG-Cu was about two orders of magnitude lower than that of bare Cu. This work highlights the special synthesized way of PECVD and superhydrophobic surface of vertical structures of graphene as coatings for various applications.

  1. Introduction need to be enriched with with graphene produced with other green roustes and I recommed citation of the following reports. Materials Letters 158 (2015) 186-189. Journal of Polymers and the Environment 26 (2018) 1072-1077

Response 3: Thank you for introducing this valuable literature. We have added the following reports in our manuscript.

  1. Comparative study need to be conducted with reported results.

Response 4: Thanks for you comment. We have added the comparative study with reported results about anticorrosion efficiency and superhydrophobic range.

  1. Conclusion need to enriched reason of anticorrosion actions.

Response 5: Thanks for your suggestion. The reason of anticorrosion actions has been enriched in conclusion. Corresponding changes have been highlighted. “……As for the protective effect of graphene and special hydrophobicity structure, the corrosion inhibition efficiency……”

Reviewer 2 Report

This review concerns manuscript titled: “In-Situ grown Vertically-oriented Graphene Coating on Copper by Plasma-enhanced CVD to Form Superhydrophobic Surface and Effectively Protect Corrosion” submitted for consideration in Nanomaterials (MDPI). The technical and scientific quality of the manuscript is sufficient. The investigated problem is relevant and the research described in the manuscript should be of interest to the Nanomaterials readership. I have a few minor technical comments and one significant criticism that I believe authors can address. 

Comments:

1. Double check grammar, spelling, and formatting. The manuscript generally reads well, but occasionally there were ungrammatical sentences, misspelled words, formatting errors, and missing or unnecessary punctuation.

2. Formatting of figure captions and Eq. 1 and 2 should be corrected.

3. Referencing panels in figure captions is confusing. Generally, (a), (b), etc. are used either at the beginning or at the end of the panel description. Here, its mixed, sometimes at the beginning and sometimes at the end. I would suggest picking one convention and using it throughout. 

4. My major criticism is that the visual inspection of corrosion for VFG-Cu sample is not sufficient to claim there is no corrosion. What if the corrosion is under the coating and not visually noticeable due to the coating. The coating clearly has a very dark color that can mask the dark green color of the possible corrosion easily. The masking may be enhanced further by the unique structure of the coating (i.e., vertically oriented graphene sheets), which may cause strong absorption or scattering. I would suggest to find a more robust method to quantify the corrosion. Perhaps cross-sectional SEM/TEM with EDX mapping of the interface to show corroded interface for the first two samples and no/little corrosion under the coating for the VFG-Cu.

Author Response

Response to Reviewer 2 Comments

  1. Double check grammar, spelling, and formatting. The manuscript generally reads well, but occasionally there were ungrammatical sentences, misspelled words, formatting errors, and missing or unnecessary punctuation.

Response 1: Thanks very much for pointing out this question. The manuscript has been doubly checked and revised. Some language and editing mistakes have been corrected and marked red in revised manuscript.

  1. Formatting of figure captions and Eq. 1 and 2 should be corrected.

Response 2: Thanks for your suggestion. The formatting of figure captions and equations have been corrected. Corresponding changes have been highlighted in manuscript.

  1. Referencing panels in figure captions is confusing. Generally, (a), (b), etc. are used either at the beginning or at the end of the panel description. Here, its mixed, sometimes at the beginning and sometimes at the end. I would suggest picking one convention and using it throughout. 

Response 3: Thanks very much for pointing out this question. The question has been revised in the manuscript. The corresponding changes have been corrected and marked in revised manuscript.

  1. My major criticism is that the visual inspection of corrosion for VFG-Cu sample is not sufficient to claim there is no corrosion. What if the corrosion is under the coating and not visually noticeable due to the coating. The coating clearly has a very dark color that can mask the dark green color of the possible corrosion easily. The masking may be enhanced further by the unique structure of the coating (i.e., vertically oriented graphene sheets), which may cause strong absorption or scattering. I would suggest to find a more robust method to quantify the corrosion. Perhaps cross-sectional SEM/TEM with EDX mapping of the interface to show corroded interface for the first two samples and no/little corrosion under the coating for the VFG-Cu.

Response 4: Thanks for your good suggestion. The test samples of cross-sectional SEM/TEM are hard to prepare and easy to harm their surfaces. Therefore, we use the electrochemical impedance spectrum measurements to robustly quantify the corrosion of Cu and VFG-Cu, which are immersed after 30 days. From the EIS spectrum, the impedance of VFG-Cu has little change and that of Cu has obvious decrease. The changes of impedances imply the different anticorrosion abilities of Cu and VFG-Cu. Relevant contents of EIS have been added in manuscript.

Reviewer 3 Report

Your manuscript sounds as a good one, however I have a problem with the abbreviations used from the very beginning. It is necessary to explain in the text what GS and GS - Cu are... Without the explanation the manuscript is incomprehensible. I have some suspicions about what is what, but the recipient/reader is not supposed to guess, but he should have a grave description of what is the subject of research. Please make necessary changes.

The sentence in the Introduction is not understable: "The VFG-Cu with superhydrophobic property as an anticorrosion coat- 62 ing, to improve the corrosion resistance of Cu" - something important is missed I think.

Author Response

Response to Reviewer 3 Comments

  1. Your manuscript sounds as a good one, however I have a problem with the abbreviations used from the very beginning. It is necessary to explain in the text what GS and GS - Cu are... Without the explanation the manuscript is incomprehensible. I have some suspicions about what is what, but the recipient/reader is not supposed to guess, but he should have a grave description of what is the subject of research. Please make necessary changes.

Response 1: Thanks for your comments. We feel apologies for this suspicion. We have added some corresponding explanation about GS and GS-Cu (generally synthesized in-plane graphene coated on Cu) in manuscript.

  1. The sentence in the Introduction is not understable: "The VFG-Cu with superhydrophobic property as an anticorrosion coating, to improve the corrosion resistance of Cu" - something important is missed I think.

Response 2: Thanks very much for pointing out this question. There is language mistake in the sentences, which has been corrected. “The VFG-Cu with superhydrophobic property could effectively improve the corrosion resistance of Cu.”

Reviewer 4 Report

This manuscript describes vertically-oriented few-layer graphene(VFG) coating on Cu by plasma-enhanced chemical vapor deposition. The authors successfully confirmed super-hydrophobicity and enhanced anticorrosion ability in VFG-coated Cu, based on water contact angle and open circuit potential curves. I recommend this paper to be published in Nanomaterials.

Author Response

Thanks for your comment.